# KI Increases Tomato Fruit Quality and Water Deficit Tolerance by Improving Antioxidant Enzyme Activity and Amino Acid Accumulation: A Priming Effect or Relief during Stress?

**DOI:** 10.3390/plants12234023

**Published:** 2023-11-29

**Authors:** Jucelino de Sousa Lima, Otávio Vitor Souza Andrade, Everton Geraldo de Morais, Gilson Gustavo Lucinda Machado, Leônidas Canuto dos Santos, Eduarda Santos de Andrade, Pedro Antônio Namorato Benevenute, Gabryel Silva Martins, Vitor L. Nascimento, Paulo Eduardo Ribeiro Marchiori, Guilherme Lopes, Eduardo Valério de Barros Vilas Boas, Luiz Roberto Guimarães Guilherme

**Affiliations:** 1Department of Biology, Institute of Natural Sciences, Federal University of Lavras (UFLA), Lavras 37203-202, MG, Brazil; jucelino.lima@estudante.ufla.br (J.d.S.L.); otaviovsandrade@gmail.com (O.V.S.A.); eduarda.andrade2@estudante.ufla.br (E.S.d.A.); vitor.nascimento@ufla.br (V.L.N.); paulo.marchiori@ufla.br (P.E.R.M.); 2Department of Soil Science, School of Agricultural Sciences, Federal University of Lavras (UFLA), Lavras 37203-202, MG, Brazil; evertonmoraislp@yahoo.com.br (E.G.d.M.); leonidas.santos2@estudante.ufla.br (L.C.d.S.); benevenutepedro@gmail.com (P.A.N.B.); gabryel.martins1@estudante.ufla.br (G.S.M.); guilherme.lopes@ufla.br (G.L.); 3Department of Food Science, School of Agricultural Science, Federal University of Lavras (UFLA), Lavras 37203-202, MG, Brazil; gilsonguluma@gmail.com (G.G.L.M.); evbvboas@ufla.br (E.V.d.B.V.B.)

**Keywords:** iodine, abiotic stress tolerance, drought, antioxidant defense, post-harvest

## Abstract

A water deficit can negatively impact fruit yield and quality, affecting critical physiological processes. Strategies to mitigate water deficits are crucial to global food security. Iodine (I) may increase the efficiency of the antioxidant system of plants, but its role against water deficits is poorly understood. This study aimed to evaluate the effectiveness of I in attenuating water deficits and improving fruit quality, investigating whether metabolic responses are derived from a “priming effect” or stress relief during water deficits. Tomato plants were exposed to different concentrations of potassium iodide (KI) via a nutrient solution and subjected to a water deficit. A water deficit in tomatoes without KI reduced their yield by 98%. However, a concentration of 100 μM of KI increased the yield under a water deficit by 28%. This condition is correlated with increased antioxidant activity, photosynthetic efficiency improvement, and malondialdehyde reduction. In addition, the concentration of 100 μM of KI promoted better fruit quality through antioxidant capacity and a decline in the maturation index. Therefore, KI can be an alternative for attenuating water deficits in tomatoes, inducing positive responses during the water deficit period while at the same time improving fruit quality.

## 1. Introduction

Drought can occur naturally, but climate change is contributing to the acceleration of this event, causing crop losses worldwide and posing a global threat to food security [1,2]. The world population is increasing and is expected to reach nine billion by 2050, consequently requiring a continuous increase in crop production [3]. However, drought is estimated to create serious plant growth problems with negative impacts on yield [4].

Water deficits are one of the principal stresses affecting plants’ physiology and biochemistry. A consequence of water deficits in plants is the indiscriminate increase in reactive oxygen species (ROS), which may lead to oxidative stress in plants, promoting the oxidation of essential molecules for plant development associated with the lipid peroxidation of membranes [5]. To minimize such adverse effects, plants adopt mechanisms of tolerance by activating enzymes of the antioxidant system, such as superoxide dismutase (SOD) and catalase (CAT) [6]. The antioxidant system is an indispensable mechanism for neutralizing ROS and, consequently, reducing cell damage caused by these molecules [7]. Lima et al. [8] and Ravello et al. [9] highlighted the importance of antioxidant enzymes in mitigating different abiotic stresses. Another strategy to deal with water deficits is the biosynthesis of osmoprotectants: small, low-molecular-weight, and non-toxic molecules, such as free amino acids like proline and some carbohydrates. The accumulation of these metabolites promotes the protection of plants against oxidative damage and improves water absorption in a water deficit [10,11].

In addition, decreases in growth and yield under a water deficit may occur due to changes in plant photosynthesis. A series of molecular, metabolic, physiological, and morphological processes are triggered in plants in response to drought conditions [12]. There is a reduction in carbon assimilation by the leaves under a water deficit, causing changes in the partition of photoassimilates and hence declines in growth and crop yield [13]. In addition, cell toxicity can cause enzymatic dysfunction, reducing the photosynthetic rate and water use efficiency [14].

Another essential aspect of the production system is consumers’ demand for quality food, highlighting its appearance, taste, nutritional value, and functional potential. However, adverse climatic conditions are often hard to control, and plants exposed to stressful environments usually lose fruit quality [15]. Quality attributes are affected by water deficits during the production of fruit-type vegetables [16]. A water deficit may increase the concentration of primary metabolites, such as organic acids and sugars, which affect flavor, as well as secondary metabolites, such as flavonoids, anthocyanins, lycopene, β-carotene, and vitamin C [17], which affect coloration, the nutritional value, shelf life, and functional potential of the fruit. Therefore, in addition to reducing yield, a water deficit may impair fruit quality, accelerating deterioration [18].

The exogenous application of mineral elements is a vital strategy to alleviate the adverse effects of water deficits on plants. Recently, several studies have recognized that the application of iodine strengthens the antioxidant capacity of soybean (*Glycine max*), lettuce (*Lactuca sativa*), and tomato (*Solanum lycopersicum* L.) plants by stimulating the activity of the main ROS detoxifying enzymes, such as SOD, ascorbate peroxidase (APX), and CAT [7,8,19,20]. Another point to highlight is the nutritional role of iodine in plants, since this element can bind covalently to at least 82 different proteins in the leaves and roots of *Arabidopsis thaliana*; its presence in micromolar concentrations in the nutrient solution resulted in increased accumulation of plant biomass and timely flowering [21].

Tomato (*S. lycopersicum* L.) requires large amounts of water. Consequently, it is negatively affected by water deficits, especially during the reproductive phase, where photosynthesis is limited, with an intensification of floral abortion and, as a result, a reduction in yield [22,23]. This vegetable is the most cultivated in the world and is one of the most nutritionally and economically important crops [24]. Among horticultural crops, it is one of the essential model species, especially the Micro-Tom cultivar, which is helpful for studies on plant tolerance to environmental stresses due to its well-known genetic profile and convenient transformation techniques [25,26].

The exogenous application of iodine to tomato plants can be an alternative for mitigating a water deficit, as demonstrated for other species [8,27]. Also, iodine has shown positive effects on improving the post-harvest quality of fruits [28]. Thus, the current study aims to ascertain whether iodine can boost the response to a water deficit in the Micro-Tom tomato cultivar while at the same time leading to the production of high-quality fruits. This study holds considerable innovation potential, as it not only addresses novel approaches to alleviating water stress in plants but also emphasizes a concurrent impact on the quality of the ultimate agricultural produce. This, in turn, promises to enhance both yield and crop quality. Given iodine’s limited exploration in agriculture, this research represents a pioneering step toward the incorporation of this element into plant nutrition programs, offering a comprehensive characterization of its role in plants and its beneficial effects for agricultural crops.

## 2. Results

### 2.1. Production

The water deficit affected (*p* < 0.05) the production and water deficit tolerance index (WDTI) of tomato plants (Figure 1). However, potassium iodide (KI) at a concentration of 100 µM increased the yield and WDTI when the plants were grown under a water deficit (Figure 1A,D). The yield was reduced by 91% in the water deficit plants at 0 and 50 µM KI concentrations compared with the control plants (optimal irrigation and 0 µM of KI) (Figure 1A). However, at the concentration of 100 µM of KI under the water deficit, the reduction in yield was 47% compared with the control. Furthermore, the concentration of 100 µM of KI increased the yield of the tomato plants by 23% compared with the other tested KI concentrations (0 and 50 µM KI) under the water deficit.

The water deficit provided a ~98% reduction in the WDTI in plants under the 0 and 50 µM KI treatments and 46% under the 100 µM KI treatment compared with the control treatment (Figure 1D). However, under the water deficit, the concentration of 100 µM of KI increased the WDTI of the plants by ~28% compared with the treatments without KI and with 50 µM of KI. There was no influence of the treatments on the number of fruits. However, the cultivation under the water deficit verified a reduction of ~71% in the dry mass per fruit compared with the control (Figure 1C).

### 2.2. Fruit Quality

The fruit quality changed depending on the water deficit and KI concentrations (*p* < 0.05). A ~3% reduction in the fruit pH occurred when the plants were subjected to a water deficit and supplemented with 50 and 100 µM of KI compared with plants without KI grown with and without a water deficit (Figure 2A). On the other hand, the titratable acidity was 65% higher in the fruits from the control treatment and those subjected to concentrations of 50 and 100 µM of KI under a water deficit when compared with the fruits under a water deficit and without KI application (Figure 2B).

Fruits from the plants treated with 50 µM of KI under a water deficit exhibited higher soluble solids content when compared with those from plants without KI treatment under a water deficit, showing an increase of 6.96%. No significant differences were observed concerning the treatments with irrigation or with 100 µM of KI under a water deficit (Figure 2C). The water deficit promoted an increase in the fruit maturation index (SS/AT), a fact mitigated by KI, in a dose-dependent manner (Figure 2D). The highest and lowest maturation index, 56.38 and 35.18, were observed in the fruits under a water deficit at 0 and 100 µM KI concentrations, respectively.

The water deficit increased the antioxidant activity of the fruits, measured through the ABTS method, by around 15% (Figure 2H). However, no differences occurred when evaluating it with the β-carotene/linoleic acid and phosphomolybdenum complex methods (Figure 2E,F). When assessing the effect of KI on a water deficit, it is noteworthy that fruits from the plants treated with 100 µM of KI had their antioxidant activity increased by around 25%, regardless of the determination method (Figure 2E,F,H). On the other hand, the dose of 50 µM of KI did not interfere with the antioxidant activity of the fruits produced by plants under a water deficit (Figure 2E,F).

As observed for antioxidant activity, the water deficit increased the total phenolic content, which was higher for the dose of 100 µM of KI (Figure 2G). Therefore, the highest concentration of total phenolics happened in the fruits harvested from the plants treated with 100 µM of KI under water deficit conditions (6233.65 mg GAE 100 g^−1^ FW) (Figure 2G). The tomato fruits produced under a water deficit had around 54% more total phenolics than the fruits not subjected to the water deficit, and the concentration of 100 µM of KI determined an increase of approximately 81% in this variable in the fruits under the water deficit.

### 2.3. Photosynthesis

The water deficit drastically reduced (*p* < 0.05) the assessed photosynthetic variables, although the concentration of 100 µM of KI minimized this effect (Figure 3). An increase of ~224% in the net photosynthesis was observed when the plants were subjected to a water deficit and exposed to a 100 µM KI (1.07 µmol CO_2_ m^−2^ s^−1^) concentration compared with those without KI under the same irrigation conditions (0.33 µmol CO_2_ m^−2^ s^−1^) (Figure 3A). In the ETR, an increase of 51% was observed in the treatment with 100 µM of KI compared with that without KI, both under a water deficit (Figure 3B). Moreover, the highest values for the net photosynthesis and ETR were observed in the control treatment (11.06 µmols CO_2_ m^−2^ s^−1^ and 83.55 µmols electrons m^−2^ s^−1^, respectively).

### 2.4. Oxidative Damage and Antioxidant Enzymes

The malondialdehyde (MDA) and H_2_O_2_ levels increased significantly (*p* < 0.05) during the water deficit (Figure 4A,B). However, the plants treated with 100 µM of KI did not change MDA content significantly before the water deficit period (Figure 4A). A ~56% reduction in MDA occurred in the plants with a treatment with 100 µM of KI (14.31 nmol MDA g^−1^ FW) compared with those without KI (31.80 nmol MDA g^−1^ FW) and 50 µM of KI (34.27 nmol MDA g^−1^ FW) during the water deficit. Regarding the concentration of H_2_O_2_, an increase of ~107% was found in the period of water deficit compared with the period before the water deficit (Figure 4B).

The antioxidant enzymatic activity also changed depending on the water deficit and KI concentrations (*p* < 0.05). For the SOD activity, an increase of ~120% was observed in the treatment without KI during the water deficit compared with the treatments before the deficit imposition (Figure 4C). Additionally, the 50 and 100 µM KI treatments increased the SOD activity during the water deficit by ~233% when compared with the same treatments before the water deficit and ~127% when compared with treatments without KI during the deficit. On the other hand, the CAT activity was only significantly influenced by the 100 µM KI treatment during the water deficit, with an increase of ~124% compared with the other treatments.

### 2.5. Osmolytes

The compatible osmolyte levels changed as a function of the treatments applied (*p* < 0.05) (Figure 5). The water deficit resulted in a significant increase in the content of total soluble sugars in the leaves, with sucrose levels rising by approximately 96% and 46%, respectively, in the treatments without and with 50 µM of KI. Notably, there was no statistically significant change in the treatment with 100 µM of KI (Figure 5A,B).

The total free amino acid levels were not affected by the water deficit, while the levels of the amino acid proline, specifically, increased as a function of stress (Figure 5C,D). KI, at concentrations of 50 and 100 µM, did not interfere with the levels of both variables under adequate water conditions. However, it caused a significant increase (*p* < 0.05) in their levels under water deficit conditions (Figure 5C,D). Potassium iodide determined an approximately 146 and 132% increase in the total free amino acids and proline, respectively, in the plants during the water deficit.

### 2.6. Multivariate Analysis

Principal component analyses (PCAs) and Pearson correlation were performed while considering the variables relating to the crop yield, fruit quality, and physiological and biochemical characteristics of the tomatoes grown under a water deficit and treated with different concentrations of KI. The first principal components (PC1 and PC2) presented 67.47% of the total data variance (Figure 6). The biplot revealed a strong relationship between the treatment without KI and the fruit ripening index and pH. These two variables correlated positively and negatively with SOD, proline, total free amino acids in leaves, titratable acidity, and ABTS in the fruits, with the respective variables being more favored by concentrations of 50 and 100 µM of KI.

The concentration of 100 µM of KI favored most of the variables analyzed, including the yield, water index tolerance, net photosynthesis, ETR, catalase, number of fruits, ABTS, phosphomolybdenum, and total phenols of the fruits. The yield, tolerance index, and net photosynthesis were positively correlated. Furthermore, the three variables mentioned above correlated positively with catalase activity. However, the yield and tolerance index correlated negatively with the content of malonaldehyde and sucrose in the leaves, which, in turn, were more favored by the concentration of 50 µM of KI. The concentration of 50 µM of KI also favored the accumulation of soluble sugars in the leaves, which is negatively correlated with the net photosynthesis. The fruit quality variables favored, in part, by the concentration of 50 µM of KI (titratable acidity and β-carotenes), in part by the concentration of 100 µM of KI (phosphomolybdenum, ABTS, and total phenols), except for β-carotene, correlated positively with net photosynthesis. However, the total phenolics, ABTS, and phosphomolybdenum correlated negatively with the total soluble sugar content in the leaves.

## 3. Discussion

Water is a vital environmental factor that affects plant growth, yield, and quality, mainly due to negative physiological and biochemical responses [29]. As in other crops, water scarcity causes a prominent inhibitory effect on tomatoes’ physiological, biochemical, and yield responses [23]. Iodine is a strategic alternative for combating abiotic stresses [8,19,30,31]. However, the effects of this element on water deficit mitigation have been little explored [8].

Our findings showed that, under water deficiency, the tomato plants had a reduction in yield, which correlated with decreases in photosynthetic variables and an increase in cell damage (MDA) (Figure 1, Figure 3, Figure 4, Figure 6 and Figure 7). Water deficits can influence redox balance and cause secondary oxidative stress with an excessive generation of reactive oxygen species (ROS) in plant cells [32,33]. Consequently, oxidative damage to biological macromolecules, such as lipids and proteins, will interfere with cellular functions and reduce plant growth and yield [33]. In photosynthetic organisms, the main sites of ROS generation occur in photosystem I and II in the chloroplast, and oxidative stress is closely related to photoinhibition and a reduction in photosynthesis [32].

Some studies have shown that the imposition of a water deficit on tomato seedlings increased the MDA content, which has been used as a biomarker of oxidative stress, also causing a decrease in ETR and net photosynthesis in tomato plants [29,34,35,36]. The results observed in the present study corroborate with those previous reports. However, applying KI at a concentration of 100 µM reduced the MDA content during the water deficit. Moreover, it improved the photosynthetic characteristics of the plants through increased antioxidant enzymatic activity, the accumulation of amino acids, and proline during the water deficit, thus providing greater yield in these plants (Figure 1, Figure 3, Figure 4, Figure 5, Figure 6 and Figure 7). These results are similar to those observed by [8], studying soybean plants subjected to water deficits and exposed to different concentrations of KI. The authors found increases in the activity of antioxidant enzymes, which in turn provided a reduction in MDA and an increase in the photosynthetic rate of these plants. Additionally, other studies with soybeans and lettuce indicated that applying I at concentrations of 20, 40, and 80 μM increased the activities of SOD, APX, and CAT [7,30].

Regarding the accumulation of amino acids and proline promoted by KI during a water deficit (Figure 5C,D), our results contrast those found in the literature. Increasing KI concentrations did not affect proline accumulation in soybean plants grown under a water deficit [8]. Additionally, Kiferle et al. [19] observed that applying iodine reduced the accumulation of these osmolytes in tomatoes subjected to saline stress. Similarly, lettuce plants subjected to saline stress and treated with iodine (KI) had reduced proline accumulation [37]. This difference in responses may have occurred due to differences in species, intensities, and exposure time to stress since the accumulation of proline and other amino acids works as a tolerance mechanism for water deficits [38]. These molecules can accumulate during environmental stresses by plants to maintain water status through osmoregulation, which prevents damage to vital plant molecules, such as DNA, proteins, and lipids, caused by ROS [39,40].

In contrast, the plants that had the highest accumulation of sucrose in the leaves (0 and 50 μM of KI under a water deficit) were the ones that had the lowest yield (Figure 1A and Figure 5B). A water deficit can inhibit plant growth, causing the accumulation of sucrose in the leaves and promoting a disturbance of the metabolic balance of sucrose [41]. Drought conditions generally increase sucrose phosphate synthase (SPS) activity and may increase sucrose accumulation [42,43,44]. Similarly to proline, the accumulation of soluble sugars (such as sucrose) acts as a source of osmolites that maintain and protect plant macromolecules and structures from stress damage, eventually increasing plant tolerance to water deficits [45,46]. However, the control plants, i.e., those not submitted to the water deficit and those treated with 100 μM of KI, which had the deleterious effects promoted by the water deficit relieved by other mechanisms, possibly translocated carbohydrates to a more substantial sink, thereby reducing floral abortion and consequently increasing yield (Figure 1A and Figure 7).

Our study indicated that the primary responses related to tolerance to water deficits promoted by KI in tomato plants were triggered during stress through relief during the deficit, dismissing a possible priming effect. During priming, a perception of stress is established, causing physiological, metabolic, and molecular changes, such as accumulating stress-responsive osmoregulatory metabolites or synthesizing protective proteins [47]. Some researchers propose that iodine can increase ROS, including H_2_O_2_, to stimulate enhanced activity of antioxidant enzymes in adverse conditions [48,49,50]. This was not observed in our study since KI did not influence the enzyme activities before the deficit, and the H_2_O_2_ content increased only when the plants were subjected to the water deficit, regardless of the KI concentrations (Figure 4).

Indeed, the water deficit promoted increased H_2_O_2_ levels (Figure 4B). Since H_2_O_2_ is toxic to cells, it must be eliminated, and CAT, which transforms H_2_O_2_ into H_2_O and O_2_, implements this. Potassium iodide at a concentration of 100 μM promoted the most significant increase in CAT (Figure 4D), which suggests its greater efficiency in controlling the deleterious effects of water deficits. Like CAT, SOD neutralizes ROS produced during oxidative processes, specifically the superoxide radical (O_2_^−^). The water deficit increased the SOD activity more pronouncedly in the fruits under the influence of KI, regardless of the dose used (Figure 4C). These results suggest, once again, the role of KI in controlling the harmful effects of water deficits, this time at the two doses tested. Malondialdehyde, an end product of lipid peroxidation, is commonly used to indicate oxidative stress. The production of free radicals increases lipid peroxidation and, consequently, MDA. The water deficit increased the production of MDA. However, this increase was significantly limited by KI at a concentration of 100 μM (Figure 4A), which demonstrates, once again, its potential to mitigate the adverse effects of this type of stress. Overall, the results point to the importance of KI in mitigating the harmful effects of free radicals and its potential as a mineral supplement in controlling or minimizing the impact of water deficits. These results are similar to those found in soybean plants treated with KI and grown under a water deficit [8].

A water deficit generally promotes the concentration of internal components (sugars, organic acids, ascorbic acid, and carotenoids, among others) due to the decrease in water content in fruits [51,52,53]. However, the water deficit did not significantly change (*p* < 0.05) the SS content of the fruits, nor their pH, although it reduced their TA. The reduction in TA can be associated with the more active consumption of organic acids due to increased metabolism caused by stress. However, these variables should not be analyzed in isolation but together since SS gives an idea of sweetness, while pH and TA reflect tomato acidity. A balance between sweetness and acidity is vital in accepting the fruit. Thus, the maturation index, also known as the ratio SS/TA, provides more relevant information than the isolated assessment of each variable. When evaluating the maturation index, it was noted that the water deficit promoted its increase. A water deficit usually accelerates fruit maturation [54], which is generally associated with an increase in the ripening index. Regardless of the dose, but more intensely at 100 μM, KI reduced the increase in the ripening index caused by the water deficit, suggesting its stress-mitigating effect and increasing the post-harvest shelf life of tomatoes.

The increase in total phenolics caused by the water deficit (Figure 2G) can be associated with their fruit concentration. Since phenolics are potent reducing agents, this increase was reflected in an increase in the antioxidant activity, measured through the ABTS+ method, which did not agree with the data obtained with the β-carotene and phosphomolybdic complex methods (Figure 2). On the other hand, the supplementation of plants under a water deficit with 100 μM of KI induced a more significant accumulation of total phenolics and improved the antioxidant activities, based on the three methods used (Figure 2E–H). These results converge with those of MDA, H_2_O_2_, SOD, and CAT, highlighting the potential of KI in mitigating the harmful effects of free radicals produced from oxidative stress. The protective effect of KI on oxidative processes minimized the consumption of phenolics as reducing agents, resulting in fruits with enhanced antioxidant activity. Our results corroborate with the results of Mejía-Ramírez et al. [55], who observed an improvement in the quality of tomatoes when evaluating the priming effect of iodine in seeds, which increased, for example, the levels of phenolic compounds and β-carotenes. Maglione et al. [56] observed that treatment with NaCl^+^ iodine improved the nutritional value and functional potential of lettuce plants in terms of bioactive compounds.

Therefore, the concentration of 100 μM of KI was the most efficient in promoting beneficial effects in conditions of stress due to a water deficit, highlighting that most of the production, quality, photosynthetic, and biochemical variables analyzed pointed to 100 μM of KI as the best treatment, as illustrated in the PCA (Figure 6). This respective concentration had the highest tolerance index to the water deficit among the plants grown under a water deficit (Figure 1D). Our findings suggest that the primary process responsible for the enhancement in tolerance to the water deficit with the application of KI was related to the increase in the production of antioxidant enzymes, with an emphasis on CAT and SOD, which promoted better protection of the photosynthetic apparatus, thus allowing for a greater yield and better fruit quality under water deficiency. Furthermore, it was found that metabolic responses were induced mainly during the water deficit.

## 4. Materials and Methods

### 4.1. Cultivation System, Experimental Design, and Treatments

Tomato cultivation was carried out in a growth chamber with an average temperature of 22 °C and a photoperiod of 12/12 h, located in the Plant Physiology Sector of the Biology Department of the Federal University of Lavras (UFLA) (21°14′45″ S, 44°59′59″ W, 920 m above sea level), southeastern Brazil. Initially, the seeds were germinated in petri dishes and then transplanted into pots. The plants were cultivated in vases with 500 g of washed sand, with one plant per pot, and fertigated once a week with 20 mL of Hoagland and Arnon [57] nutrient solution for ten weeks. The KI concentrations applied weekly were 0, 50, and 100 µM. Therefore, the total amounts of I applied were approximately 1.27 and 2.53 mg of I kg^−1^ of substrate in the treatments of 50 and 100 μM of KI, respectively.

The treatments added to the pots were arranged in a completely randomized design with four replicates and two pots per experimental plot, so there was the possibility of collecting plants for evaluations in two moments (before and during water deficit) and evaluating yield. Therefore, the experiment had 32 experimental units, with whole plants collected before the water deficit (one day before), to assess possible priming effects on the antioxidant system, oxidative stress, and osmolyte content. In addition, one leaf per plant was collected during the deficit to evaluate the effects of the deficit (one day before rehydration). On both collection occasions, the samples were homogenized, forming a composite sample from the two pots of each treatment to minimize variations per plant. The experiment lasted 90 days until the species’ reproductive cycle ended. The treatments used are shown in Table 1 below.

### 4.2. Production and Water Deficit Tolerance Index

The fresh weight of the fruits on each plant was considered for the yield assessment. The number of fruits per plant and the average weight of a fruit were also evaluated. Furthermore, the WDTI was calculated according to [58] using the following equation:$$WDTI=\left(\frac{yield\;of\;reference\;plants}{yield\;of\;the\;other\;treatments}\right)\times 100$$

### 4.3. Fruit Quality

The fruit pulp was crushed in water in a 1:3 ratio (m/v); the homogenate was filtered in an organza cloth; and the filtrate was used for the pH, titratable acidity (TA), and soluble solids (SS) determinations. The pH was determined using a Tecnal^®^ pH meter, previously calibrated using buffer solutions (pH 4.0 and 7.0). The titratable acidity was determined via titration with a 0.01 N sodium hydroxide (NaOH) solution, using phenolphthalein as an indicator, according to the AOAC [59]. The results were expressed in mg of citric acid 100 g^−1^ of the sample. The soluble solids were determined in an ATA-GO PR-100 digital refractometer (Tokyo, Japan) with automatic temperature adjustment, and the results were expressed in %, as described by the AOAC [59]. The SS/TA ratio, the maturation index, was calculated.

Obtaining extracts for the quantification of total phenolic compounds and antioxidant activity was carried out according to the procedure of Rufino et al. [60]. Briefly, 0.5 g of a sample was homogenized, along with 4 mL of 50% methanol and 4 mL of 70% acetone, in a centrifuge tube for 30 min on a shaking table, protected from light. Then, the tubes were taken to an ultrasonic bath for 30 min, and the homogenate was filtered through filter paper (qualitative filter paper, 15 cm in diameter, Unifil^®^, Carvalhaes Produtos para Laboratorio LTDA, Rio Grande do Sul, Brazil). The filtrate was transferred to a 10 mL volumetric flask, and the volume was topped up with distilled water. The obtained extracts were placed in amber bottles and stored in a freezer until the total phenolics and antioxidant activity were analyzed.

The total phenolic compounds were determined through the method described by Medina [61], using Fast Blue BB, with some adaptations. Fifty µL of the extract was mixed with 200 µL of distilled water, 25 µL of Fast Blue reagent (0.1%, *v*/*v*), and 25 µL of sodium hydroxide (5%, *w*/*v*), and the absorbance was measured at 420 nm after 1.5 h of incubation in the dark. All measurements were performed in triplicate using a microplate reader. The results were reported as gallic acid equivalents in mg 100 g^−1^ fresh sample mass (mg GAE 100 g^−1^ FM).

The antioxidant activity was determined with the β-carotene/linoleic acid, ABTS+, and phosphomolybdenum complex. The determination of the antioxidant activity through the β-carotene/linoleic acid method was based on β-carotene oxidation (discoloration) induced by the oxidative degradation products of linoleic acid ([60], with modifications). Solutions were prepared by mixing 270 μL of β-carotene/linoleic acid system solution and 30 μL of extract into each well of a 96-well, flat-bottomed microplate. The mixture was kept in a water bath at 40 °C, and the readings were performed at 470 nm after 2 h. The results were expressed as a percentage of oxidation inhibition.

Determining the antioxidant activity through the ABTS+ method was based on capturing the ABTS+ radical with an antioxidant. Briefly, the ABTS+ solution was prepared by reacting diammonium salt 2,2’-azinobis (3-ethylbenzothiazoline-6-sulfonic acid) at a concentration of 7 mmol L^−1^ with potassium persulfate 2.45 mmol/L at room temperature for 16 h. The obtained solution was then diluted with ethanol until an absorbance of 0.70 ± 0.05 at 734 nm was reached. Aliquots of 5 μL of sample extracts were pipetted into a 96-well, flat-bottomed microplate. An aliquot of 295 μL of solution (ABTS+) was added to each well. After 6 min of reaction time in the dark, the absorbance was measured at 734 nm. The results of antioxidant activity were expressed in % ABTS Reduction, according to the equation below [62].
% ABTS Reduction = (ABS control − ABS sample)/(ABS control) × 100

Antioxidant determination through the phosphomolybdenum complex, based on the reduction of Mo^6+^ to Mo^5+^, was performed as described by Prieto et al. [63], with some adaptations. Fifty µL of the sample extract, 450 µL of distilled water, and 1.5 mL of the phosmolybdenum complex were pipetted into tubes with screw caps, which were closed, shaken, taken to a water bath at 95 °C for 90 min, and cooled in an ice bath. The absorbance reading was performed using a spectrophotometer at 695 nm, and the results were expressed in mg ascorbic acid 100 g^−1^ fresh matter.

### 4.4. Net Photosynthesis and Electron Transport Rate (ETR)

The net photosynthesis and electron transport rate (ETR) analyses were performed on the last day of the water deficit. The net photosynthesis was measured using an infrared gas exchange analyzer (IRGA, model LICOR 6400, Li-COR Biosciences, Lincoln, NE, USA). Data collection was performed between 8 a.m. and 10 a.m. Atmospheric CO_2_ inside the leaf chamber was maintained at 400 μmols CO_2_ mol air^−1^, irradiance at 1500 μmols photons m^−2^ s^−1^, and leaf temperature at 25 °C. The pre-established minimum time for the stabilization of the readings was 120 s.

### 4.5. H_2_O_2_ and MDA Content

A mass of 0.2 g of fresh material was collected and macerated in a mortar with liquid nitrogen, homogenized in 1500 μL of trichloroacetic acid (TCA) 0.1%, and centrifuged at 12,000× *g* for 15 min at 4 °C. The H_2_O_2_ content was determined through a reaction with potassium iodide (KI), according to Alexieva et al. [64]. Readings were performed in a spectrophotometer at 390 nm. The amount of H_2_O_2_ was expressed in μmol g^−1^ of fresh mass. The quantification of MDA was carried out through the reaction of TBA (2-thiobarbituric acid) with the final products of the lipid peroxidation process and the readings taken in a spectrophotometer at 535 and 600 nm, obtaining MDA values, which were calculated according to the equation described by Heath and Packer [65]. The amount of MDA was expressed in nmol g^−1^ of fresh mass.

### 4.6. Antioxidant Enzymatic Activity

For the evaluation of the enzymatic activity of SOD, and CAT, 0.2 g of fresh material was macerated in liquid nitrogen with the subsequent addition of 1.5 mL of a buffered solution (0.1 mol L^−1^ of potassium phosphate pH (7.8), 0.1 mol L^−1^ of EDTA (pH 7.0), 0.5 mol L^−1^ of DTT, 0.1 mol L^−1^ PMSF, 1 mmol L^−1^ of ascorbic acid, and 22 mg of PVPP). After the suspension was centrifuged at 13,000× *g* for 10 min at 4 °C, the supernatant was collected for analysis in a microplate reader.

The superoxide dismutase activity was determined by quantifying the inhibition of photoreduction in nitroblue tetrazolium (NBT), following the protocol developed by [66]. Sample absorbances were recorded at 560 nm. As for CAT activity, the reaction solution was made from a mixture of 30 mM of H_2_O_2_, an aliquot of the enzymatic extract (supernatant), and sodium phosphate buffer (100 mM and pH 6.0). The absorbance was observed in the time scan (0–60 s) at 240 nm [67].

The enzymatic extraction method determined the total soluble protein content to calculate the specific activity of antioxidant enzymes. The microplates initially received 294 μL of Bradford [68] solution in a 1:5 dilution of the reagent and aliquots of the enzymatic extract. Readings were performed at a wavelength of 595 nm, and the results were obtained from a calibration curve with BSA.

### 4.7. Compatible Osmolytes

The proline content was extracted according to the methodology proposed by Bates et al. [69]. Initially, 0.05 g of plant material was macerated in 10 mL of 3% (*w*/*v*) sulfosalicylic acid. Subsequently, quantification was carried out following the method described by Carillo et al. [70]. A reaction mixture was prepared, comprising 3 mL and consisting of 1 mL of the extract obtained after maceration, 1 mL of ninhydrin acid, and 1 mL of glacial acetic acid at 99.5% (*v*/*v*). The mixture was stirred and subsequently subjected to heating in equipment with water at 100 °C for 60 min. After this time, the samples were transferred to ice to stop the reaction for 10 min. The supernatant was collected, its absorbance was read at 520 nm, and the results were expressed in μmol proline g^−1^ of fresh mass.

The extraction total soluble sugars and total free amino acids was performed according to Zanandrea et al. [71], using 0.05 mg of dry weight and homogenized in 5 mL of potassium phosphate buffer 100 mM (pH 7.0) and then placed in a water bath for 30 min at 40 °C. Homogenate was centrifuged at 5000 g for 10 min and the supernatant was collected. The process was repeated twice, and supernatants were combined. Total soluble sugars and sucrose were quantified as described by Dische [72]. The determination of was based on the colorimetric method using ninhydrin, established by Yemm and Coccking [73].

### 4.8. Statistical Analysis

The data were subjected to Shapiro–Wilk normality tests and Barlett’s homogeneity of variance test. When the assumptions were met, they were subjected to a two-way ANOVA with the post-hoc Tukey test. The data were presented in bar graphs. When the data did not show normality or variance homogeneity, a rank transformation of the data was performed [74], and the data were represented in a boxplot so that it was possible to observe the data dispersion better. The PCA was used to analyze the multivariate correlation between all morphophysiological and biochemical variables and the treatment conditions. All statistical analyses and graphs were made under the R software environment using the tidyverse [75], multcomp [76], and rstatix [77] packages.

## 5. Conclusions

Exposure to iodine (as KI) increased tomato plants’ water deficit tolerance through increased antioxidant enzymatic activity and the accumulation of amino acids. These responses were induced during a water deficit, providing greater photosynthetic efficiency and yield. In addition, iodine (I) promoted better fruit quality by increasing the antioxidant capacity of phenolic compounds and reducing the maturation index. Based on the insights gained for all variables in conjunction with the PCA, the treatment with 100 µM of KI offered the best performance. Based on the results, it was also evident that most responses induced by iodine occurred during periods of water deficit. This observation strongly suggests that iodine elicits relief responses rather than priming effects. Additional studies should be carried out with the application of different sources and forms of I to identify the best strategies to mitigate water deficiency under commercial production conditions, where the use of iodine, along with other products, would be evaluated to assess possible interactions of I with other elements/components in the soil. Also, an increasing sampling frequency should be considered for a better understanding of metabolic responses and the induction moments of the responses promoted by iodine under water deficits. Finally, the beneficial effects of I on the post-harvest quality of fruits, as well as on other relevant physical–chemical and nutritional characteristics, should be assessed in future investigations.

## Figures and Tables

**Figure 1 plants-12-04023-f001:**
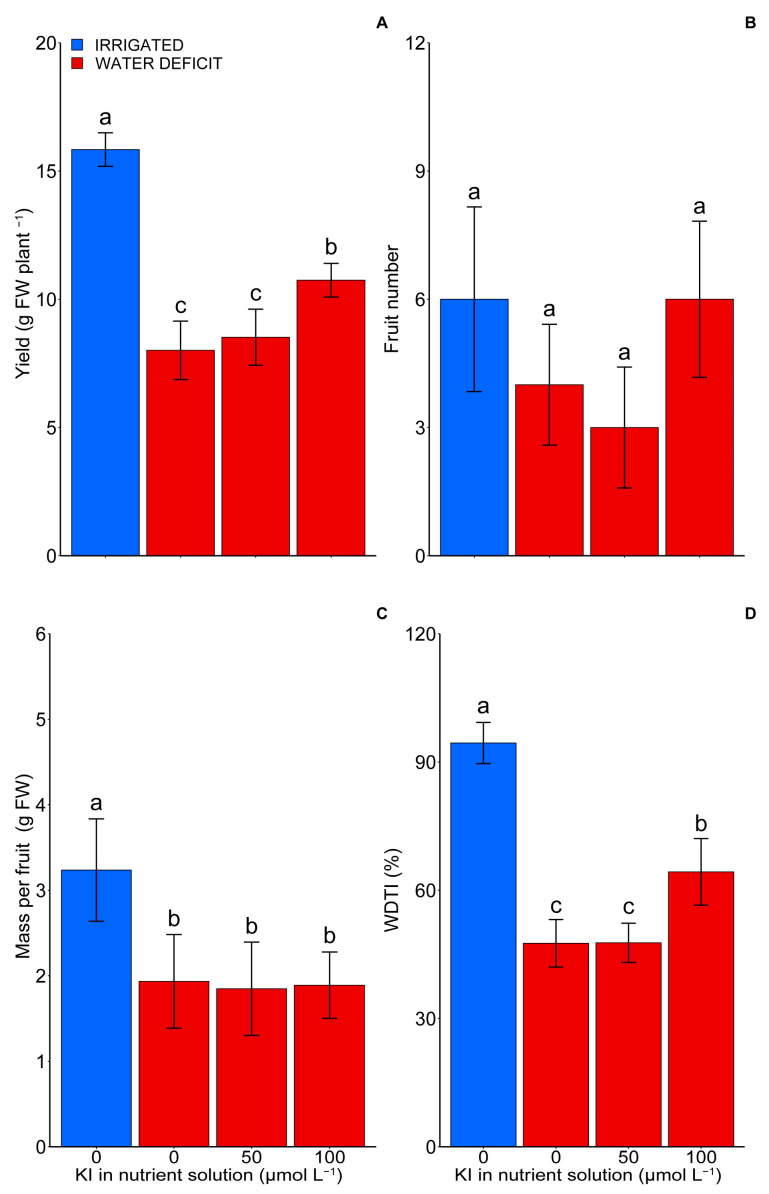
Effect of KI on yield (**A**), number of fruits (**B**), mass per fruit (**C**), and water deficit tolerance index—WDTI—(**D**) in tomato plants under water deficit or not. The control treatment (without water deficit) corresponds to the blue bars (first bar), while the second, third, and fourth bars, in red, represent the water deficit condition and the treatments with 0, 50, and 100 µmol L^−1^ of KI, respectively. Values are average ± SD (n = 4). The same letters indicate no significant difference (*p* > 0.05) in the Tukey test. The fruits were harvested 90 days after the seeds’ germination, and all the clusters were collected.

**Figure 2 plants-12-04023-f002:**
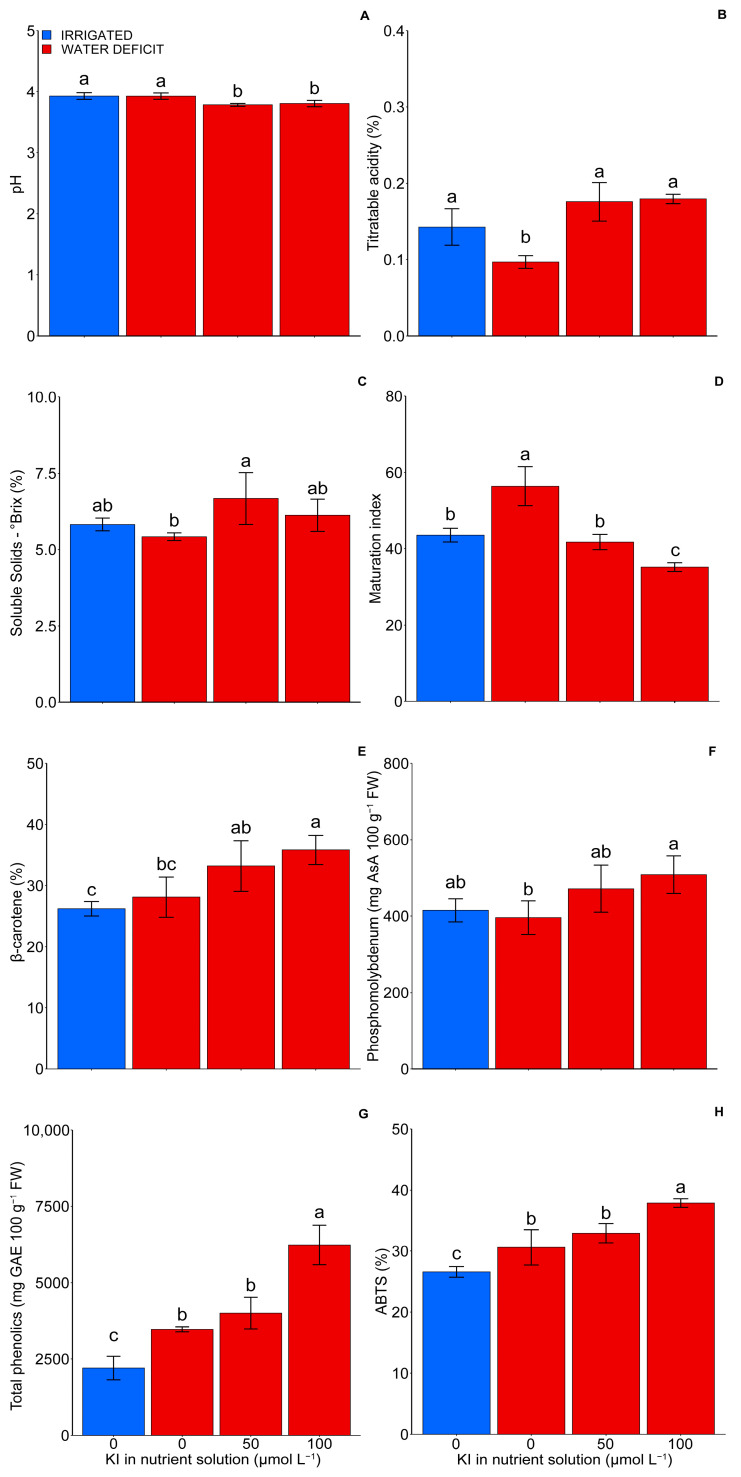
Effect of KI on pH (**A**), titratable acidity (**B**), soluble solids (**C**), maturation index (**D**), β-carotene, (**E**) phosphomolybdenum (**F**), total phenolics (**G**), and ABTS (**H**) in tomato fruits under water deficit or not. The control treatment (without water deficit) corresponds to the blue bars (first bar), while the second, third, and fourth bars, in red, represent the water deficit condition and the treatments 0, 50, and 100 µmol L^−1^ of KI, respectively. Values are average ± SD (n = 4). The same letters indicate no significant difference (*p* > 0.05) in the Tukey test.

**Figure 3 plants-12-04023-f003:**
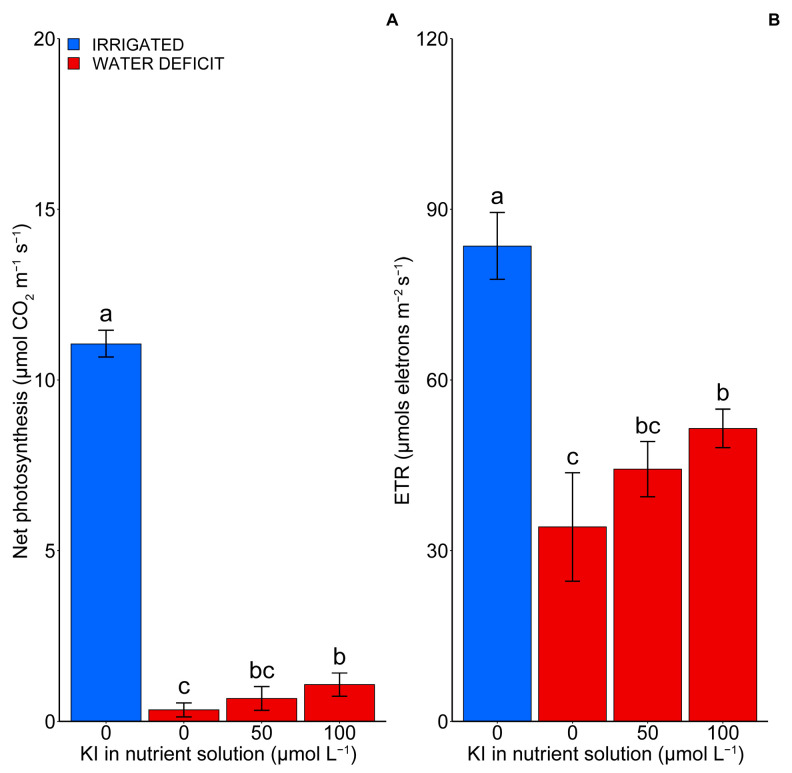
Effect of KI on net photosynthesis (**A**) and electron transport efficiency rate (**B**) in tomato plants under water deficit or not. The control treatment (without water deficit) corresponds to the blue bars (first bar), while the second, third, and fourth bars, in red, represent the water deficit condition and the treatments 0, 50, and 100 µmol L^−1^ of KI, respectively. Values are average ± SD (n = 4). The same letters indicate no significant difference (*p* > 0.05) in the Tukey test.

**Figure 4 plants-12-04023-f004:**
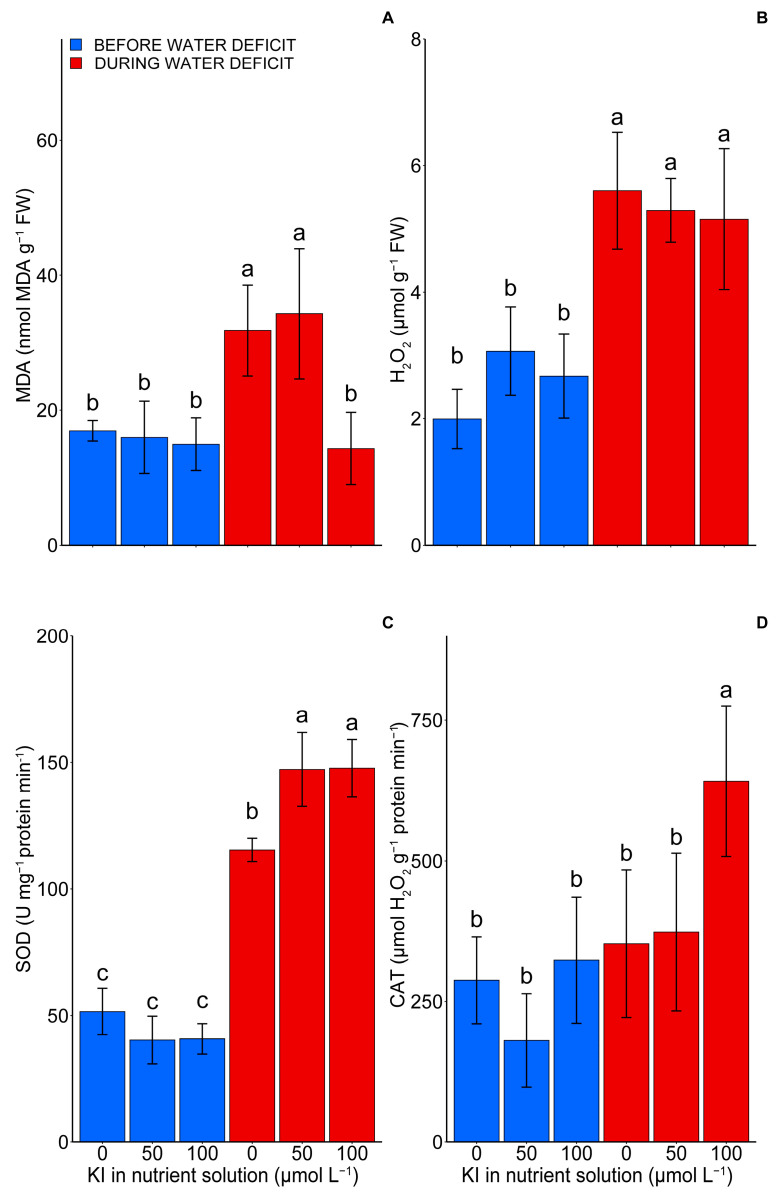
Effect of KI on malondialdehyde content (**A**), hydrogen peroxide (**B**), superoxide dismutase activity (**C**), and catalase activity (**D**) in tomato plants before and during water deficit. The first, second, and third blue bars represent the condition before the deficit and the treatments 0, 50, and 100 µmol L^−1^ of KI, respectively. In contrast, the first, second, and third red bars represent the condition during the deficit and the treatments 0, 50, and 100 µmol L^−1^ of KI, respectively. Values are average ± SD (n = 4). The same letters indicate no significant difference (*p* > 0.05) in the Tukey test.

**Figure 5 plants-12-04023-f005:**
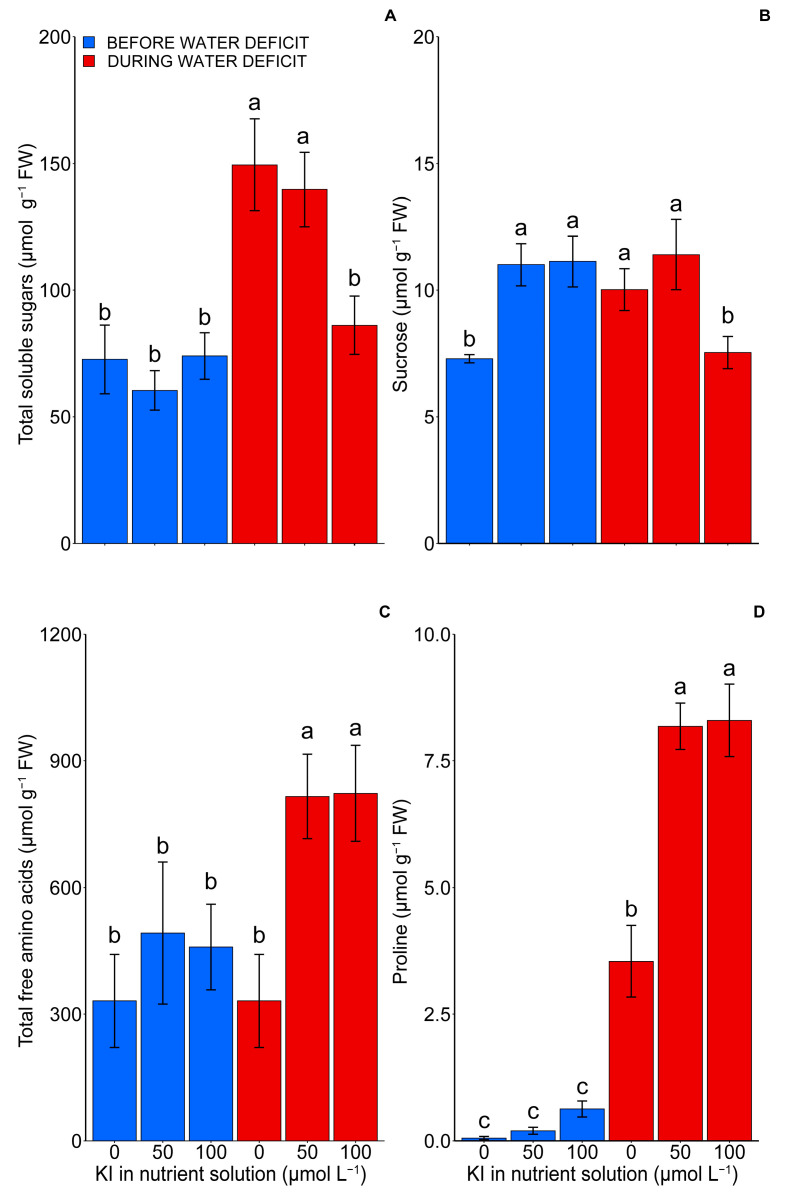
Effect of KI on total soluble sugars (**A**), sucrose (**B**), total free amino acids (**C**), and proline (**D**) in tomato plants before and during water deficit. The first, second, and third blue bars represent the condition before the deficit and the treatments 0, 50, and 100 µmol L^−1^ of KI, respectively. In contrast, the first, second, and third red bars represent the condition during the deficit and the treatments 0, 50, and 100 µmol L^−1^ of KI, respectively. Values are average ± SD (n = 4). The same letters indicate no significant difference (*p* > 0.05), and different letters represent a significant difference (*p* < 0.05) in the Tukey test.

**Figure 6 plants-12-04023-f006:**
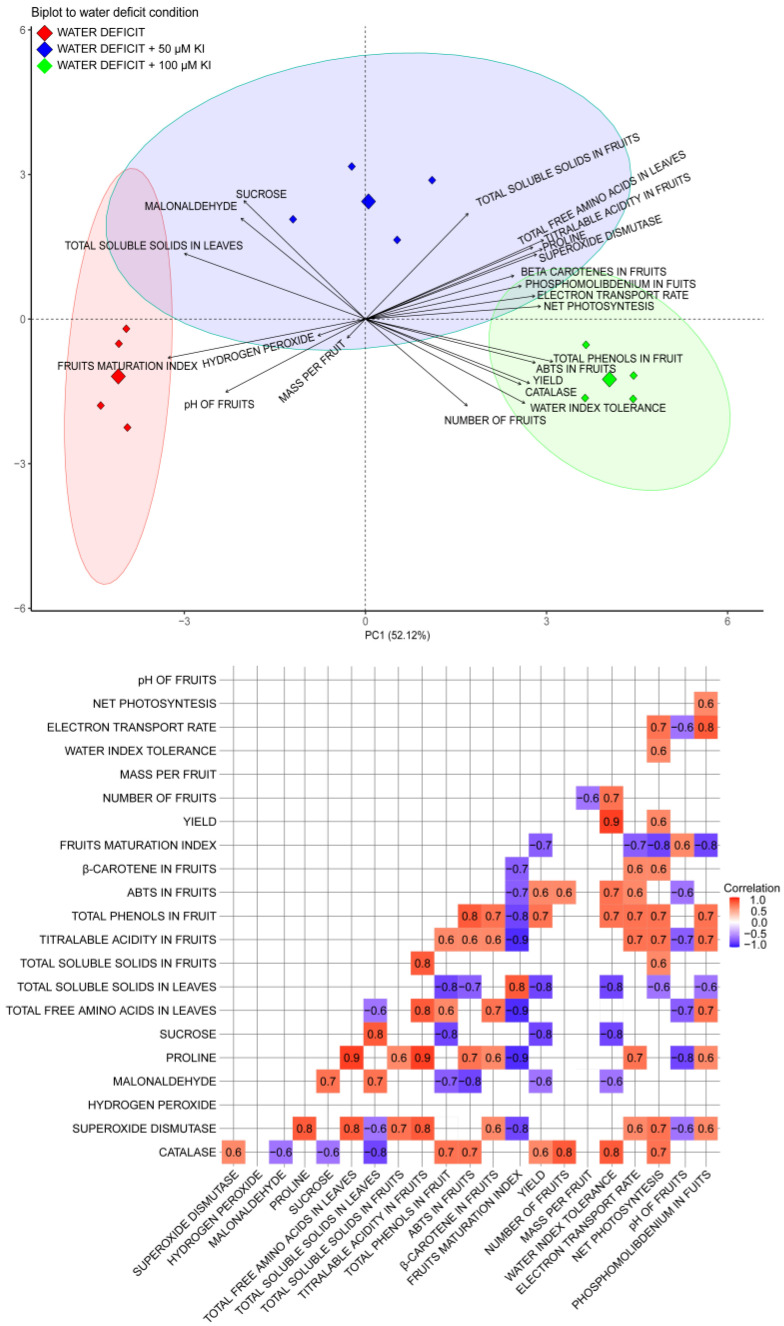
Principal component analysis and Pearson correlation using tomato plants’ production, fruit quality, and physiological and biochemical characteristics data in response to water deficit conditions and KI application. Significant correlation coefficients (*p* < 0.05) are indicated by bold numbers, with positive and negative correlations distinguished by red and blue, respectively. White boxes indicate non-significance without numbers.

**Figure 7 plants-12-04023-f007:**
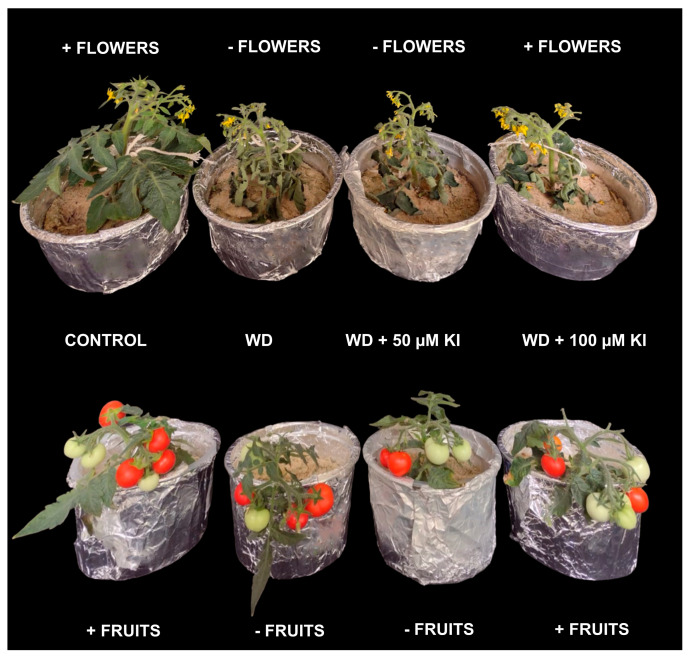
Representation of treatments in the flowering phase under water deficit and in the reproductive phase. Control: plants under optimal irrigation and without KI application; WD: plants subjected to water deficit; WD + 50 µM KI: plants subjected to water deficit and a concentration of 50 μM of KI; and WD + 100 µM KI: plants subjected to water deficit and a concentration of 100 μM of KI.

**Table 1 plants-12-04023-t001:** Representation of treatments applied in the experiment.

Treatments	Description
Treatment 1 (Control)	Irrigation + 0 μM KI
Treatment 2	Water Deficit + 0 μM KI
Treatment 3	Water Deficit + 50 μM KI
Treatment 4	Water Deficit + 100 μM KI

## Data Availability

Data are contained within the article.

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
