# Peer review of "KI Increases Tomato Fruit Quality and Water Deficit Tolerance by Improving Antioxidant Enzyme Activity and Amino Acid Accumulation: A Priming Effect or Relief during Stress?"

_plants, 2023, doi:10.3390/plants12234023_

Round 1
Reviewer 1 Report
Comments and Suggestions for Authors
Title: KI increases tomato fruit quality and water deficit tolerance by improving antioxidant enzyme activity and amino acid accumulation: a priming effect or relief during stress?
Tomato is one of the primary vegetables and its production are severely impaired due to occurrence of drought stress/ water deficit stress. Further this stress is complicated and others also happens simultaneous. This study established that iodine (as KI) application increased tomato plants' water deficit tolerance through increased antioxidant enzymatic activity, accumulation of amino acids, greater photosynthetic efficiency, yield, better fruit quality, increasing the antioxidant capacity of phenolic compounds and reducing the maturation index. Also established the fact that KI application act as relief during stress.
However, I have some suggestions to improve the presentation of this manuscript.
Abstract: Concise covering a background and methodology performed and significant results obtained in this study.
Line 21, water deficit in tomato.
Line 25, the concentration of 100 μM KI promoted
Introduction: Detailed account on importance of the study
Line 42, ance by the activation of enzymes of the antioxidant system
Line 45, Lima et al. [8] and Ravello et al. [9] highlighted…
Result section: well presented.
I could see some calculation error. Please check % reduction and percent improvement throughout the result section. Formula used for calculation must be given in the supplementary file.
Figure 1B, Fruit number and Fig. 1C fruit mass is very low for 90 days grown plants
Fig. 2C, not properly explained in the result section
Fig. 3A, Please check the calculation
Fig. 4B and Fig 4C, Please check the calculation
Fig 5A and 5B: Please check the calculation
This section needs improvement.
Discussion: Well discussed. Following references may cited as
Line 251-252, However, the effects of this element against water deficit mitigation have been little explored [8].
Line 284-289, KI concentrations did not affect proline accumulation in soybean plants grown under water deficit [8]. Additionally, Kiferle et al. [19] observed that applying iodine reduced the accumulation of these osmolytes in tomatoes subjected to saline stress. Similarly, lettuce plants subjected to saline stress and treated with iodine, observed that applying KI reduced proline accumulation [38].
Line 360-364, Our results corroborate with results of Mejía-Ramírez et al. [56], who observed an improvement in the quality of tomatoes when evaluating the priming effect of iodine in seeds, increasing, for example, the levels of phenolic compounds and β-carotenes. Maglione et al. [57] observed that treatment with NaCl + iodine improved the nutritional value and functional potential of lettuce in terms of bioactive compounds.
Materials and methods: Given in detail and easy to follow by the readers.
Line 379, include relative humidity and light intensity used growth chamber
Line 419, activity was carried out according to the procedure of Rufino et al. [61].
Line 427, Total phenolic compounds were determined by the method described by Medina [62],
Line 453, was performed as described by Prieto et al. [64],
Line 471, iodide (KI), according to Alexieva et al. [65].
Line 476, described by Heath and Packer [66].
Line 496, The proline content was determined according to the methodology proposed by Bates et al. [70].
Conclusion: Appropriate based on the results

Reviewer 2 Report
Comments and Suggestions for Authors
Lima et al. assessed how KI affects water deficit tolerance and fruit quality in tomatoes. This is an interesting study that provides another case that idione is involved in stress response in plants. However, there are several major issues that need to be addressed.
1. The language should be extensively revised.
2. the authors ask “……a priming effect or relief during stress?” This is a critical question, but what’s the answer to it? The authors did not provide information in Discussion or Conclusion.
3. Lines 174-175, “……plants treated with 100 µM KI did not change MDA content significantly during the water deficit period”, this not correct, as shown in Fig. 4A, 100 µM KI obviously decreased MDA content under stress conditions.
4. Line 177-178, “an increase of ~107% was found in the period of water deficit compared to the period before the water deficit” The description is not accurate, Which KI treatment were the authors referring to?
5. Lines198-199, “The water deficit promoted in the leaves increases the total soluble sugar and sucrose content by around 96 and 46%, respectively, which KI mitigated at 100 µM“. The description is confusing, please modify this sentence.
6. Line 252, “……highlighting the research of [8],” It appears that some information was missed.
7. The subtitles in Results section should not be a single word or just two words.
Comments on the Quality of English LanguageThe language should be extensively revised.
Reviewer 3 Report
Comments and Suggestions for Authors
1. The manuscript presents a study of KI on tomato fruit quality under water deficit conditions. Considering that KI against water deficit is poorly understood, I think that the manuscript presents interesting findings and most of the research objectives have been achieved. Overall, the manuscript is worthy of publication. However, I have several concerns that should be addressed before the paper could be published.
2. In the Discussion part, it is better to discuss how the effect and meaning of KI with higher concentrations (such as 150 uM) on tomato fruit quality under water deficit conditions will be. What’s more, it is better for the authors to supplement the role of the tomato cultivars (tolerant or sensitive) in evaluating the effect of KI on the tomato fruit quality under water deficit conditions. In addition, it is better for authors to discuss and show the application potential of KI with 100 uM in improving the fruit quality under water deficit conditions.
Round 2
Reviewer 1 Report
Comments and Suggestions for Authors
Title: KI increases tomato fruit quality and water deficit tolerance by improving antioxidant enzyme activity and amino acid accumulation: a priming effect or relief during stress?
Thank you for clarifying the points raised in the review process.
Your points are agreed and your answers are meticulous.

Author Response
Dear Reviewer, we extend our gratitude for your valuable suggestions. Your insights have greatly contributed to the enhancement of our manuscript.
Reviewer 2 Report
Comments and Suggestions for Authors
The authors addressed most of my concerns.
Comments on the Quality of English LanguageThe authors should carefully check the language.
Author Response
Dear Reviewer, we extend our gratitude for your valuable suggestions. Your insights have greatly contributed to the enhancement of our manuscript. We have diligently addressed the proposed revisions.